# A Diversity of Approaches to Visual Impact Assessment

**James F. Palmer [1,2]** 

1   Scenic Quality Consultants, Burlington, VT 05408, USA; palmer.jf@gmail.com
2   T.J. Boyle Associates, Burlington, VT 05401, USA

**Abstract:** Within the broad field of landscape assessment, scholarship focused on visual impacts accounted for a very small percentage of peer-reviewed journal articles. There has been even less interest in reviewing the methods being employed to conduct visual impact assessments (VIA) as part of the development permitting process. The purpose of this review has been to show that VIA is not a uniform concept and includes a rich diversity of methods. Six diverse methods are described in some detail, including the assessment instruments, and then comments on the apparent dimensions of this diversity are made. The discussion compares the objectives, criteria, indicators, and standards used by these methods. It also considers whether the VIA is conducted using quantitative or qualitative measurements, professional or public assessors, and a geographic information system (GIS) or key observation point (KOP) framework. It concludes with a discussion of recommendations concerning identifying a gold standard against which to evaluate the VIA methods, the use of multiple methods in a VIA, and the need for research that evaluates the validity and reliability of tools and indicators as they are used in VIA.

**Keywords:** contrast rating; guidelines for landscape and visual impact assessment; Queensland scenic amenity method; the Spanish Method; Maine Wind Energy Act

## 1. Introduction

The practice of assessing environmental impacts has spread around the globe, and by the end of the century, Sadler [1] reported that "more than 100 countries and organizations" had adopted the use of environmental assessments, making it "one of the more successful policy innovations of the 20th Century". One aspect of Sadler's study developed country status reports based on a detailed survey sent to environmental assessment agencies. Twenty responses provided sufficient detail to permit a comparative analysis, an aspect of which was identifications of the factors that can be considered in the environmental assessment. Visual impacts were not identified as one of the primary factors and were placed into the "other" category along with "cultural–historical heritage, archaeology, visual impact, material assets, land use by indigenous peoples, community structure, and landscape". Collectively, these other factors were only considered by 9 of the 20 represented countries. Recognition of visual change as an important component in environmental impact assessment began in the Unites States and the United Kingdom, followed by the European Union and the former members of the Commonwealth of Nations.

In response to the environmental movement of the 1960s, the U.S. Congress passed the National Environmental Policy Act [2], requiring Federal agencies to prepare a detailed statement describing the environmental impacts of their actions. From its beginning, one of NEPA's purposes was to "assure . . . aesthetically and culturally pleasing surroundings". This is in keeping with a long U.S. tradition of protecting scenery. For instance, the National Park Service [3] was established "to conserve the scenery and the natural and historic objects and the wild life therein and to provide for the enjoyment of the same in such manner and by such means as will leave them unimpaired for the enjoyment of future generations". This recognition of aesthetics or scenic impacts led to the inclusion of a visual impact assessment as part of the broader environmental impact assessment. Smardon and

Karp [4] have reviewed the ways that the individual states have regulated impacts on scenic quality and aesthetics.

In 1985, the European Commission established the Environmental Impact Assessment Directive, which included identifying "the direct and indirect effects of a project on . . . the landscape". Placing a priority on the landscape rather than the scenery is characteristic of the European approach, as embodied in the European Landscape Convention [5]. It defines landscape as "an area, as perceived by people, whose character is the result of the action and interaction of natural and/or human factors", which clearly involves its appearance. Fairclough and colleagues [6] have assembled reviews of current approaches to the characterization and assessment of landscape in Europe and other parts of the world. There was no explicit mention of visual concerns until the 2014 amendments to the EIA Directive, which stated that "it is important to address the visual impact of projects, namely the change in the appearance or view of the built or natural landscape and urban areas, in environmental impact assessments".

However, these laws only require that visual impacts be considered but do not normally specify the methods by which the change should be assessed. While there have been reviews of the visual impact assessment literature [7,8], they have focused on the research literature, particularly the literature on the perception and assessment of existing landscapes, and not the methods being used to conduct visual impact assessments (VIA). This is an important distinction. For instance, the landscape perception research generally compares very different landscape views, while VIAs must consider the same view with and without a proposed project. An objective of landscape perception research is to identify landscape attributes (e.g., relief, naturalness) that correlate with a rating of the view's scenic quality. However, this is not suitable for understanding VIA, which needs to determine whether the difference between the existing and proposed condition is significant or unreasonably adverse.

This essay initiates a discussion on how VIAs are being conducted and suggests a number of considerations for improving them.

## 2. Methods

Six well-documented and recognized methods are selected to demonstrate the diversity of approaches used to conduct VIA. There is no attempt to represent how frequently they are employed, though that would be an interesting research question. Their selection is based on the author's 40-plus years of experience as an academic and practitioner in the field. A not insignificant but necessary limitation is that documentation must be available in English on the internet. The six methods are:

1.  BLM Contrast Rating System (BLM);
2.  Berkeley Contrast Rating (UCB);
3.  Guidelines for Landscape and Visual Impact Assessment (GLVIA);
4.  Queensland Scenic Amenity Methodology (SAM);
5.  The Spanish Method (SP2);
6.  Maine Wind Energy Act (WEA).

The following summary for each method includes the background to its creation, a description of the methodological approach, and a list of some of its distinctive characteristics.

- What is it objective or purpose of the method?
- What are the criteria or conditions that are used to assess the objective?
- What are indicators that are used to describe or measure the criteria?
- What is the standard or threshold used to determine impact significance?
- Is measurement qualitative or quantitative?
- Is the process based on a professional appraisal or public assessment?
- Is it based primarily on a GIS to analyze the whole study area or simulations at a few selected key observation points (KOPs)?

## 3. Results

### *3.1. BLM Contrast Rating System*

#### 3.1.1. Background

In the U.S., the Forest Service (FS) led the way in developing a systematic approach to managing scenery as a natural resource in the late 1960s and 1970s [9] (Appendix G). However, this did not include a procedure to assess visual impacts. By 1975, the Bureau of Land Management (BLM) also became involved in scenery management, including a concern about how to systematically evaluate visual change in the landscape. The approach taken by the BLM evolved from discussions among FS and BLM landscape architects trying to understand how people "see" the landscape and how to talk about it between the agencies [10,11]. In particular, they believed that visual impacts resulted from an activity's visual contrast with its surroundings. Their discussion was grounded in their landscape architecture education, which included a Modernist approach to design education and Gestalt principles of perception. The result was an approach that deconstructs the view into its basic elements-form, line, color, and texture-as well as principles resulting in perceived visual dominance and variable factors that influence views. At the time, the BLM had less than 20 landscape architects, and they envisioned a procedure that non-design professionals could implement with a modicum of training. By employing the procedure, staff could determine where a proposed action's potential visual effects required the attention of a landscape architect to plan for appropriate mitigation. The BLM approach does not quantitatively measure impact; it describes the source of visual contrasts to determine if management objectives are met and to guide visual mitigation, if necessary.

#### 3.1.2. BLM Contrast Rating

The BLM's visual resource management (VRM) conducts a systematic visual resource inventory that maps visual resource classes and assigns them visual resource objectives [12]. The procedure involves mapping the management area into units of high, medium, and low scenic quality based on an evaluation of landform, vegetation, water, color, adjacent scenery, scarcity, and cultural modifications. The area is also mapped into units of high, medium, and low sensitivity based on the type of users, amount of use, public interest, adjacent land uses, special designations, and other factors. Some of these attributes may have quantitative thresholds, while others are based on professional judgment.

VRM classes are determined using a table that combines scenic quality, visual sensitivity, and distance from special areas (e.g., wilderness), travel routes, or observation points. The VRM class determines the visual management objectives for an area.

As part of a VIA, contrast ratings are used to describe the visual effect of a proposed action and determine whether it is consistent with the VRM class objectives. It is conducted at key observation points (KOPs), which are described as "the most critical viewpoints . . . usually along commonly traveled routes or at other likely observation points" [13].

The assessment is made in the field at the KOP, and while not required, a visual simulation is normally available to help understand how the view will change. Contrast ratings are recorded on a field sheet, shown in Figure 1, that systematically guides the assessment. The form, line, color, and texture of the existing landform/water body, vegetation, and structures are described. These attributes are also described for how the view will appear if the activity is implemented.

Form 8400-4

**UNITED STATES**
**DEPARTMENT OF THE INTERIOR**
**BUREAU OF LAND MANAGEMENT**
**VISUAL CONTRAST RATING WORKSHEET**

| | |
|---|---|
| Date: | Aug 15, 1985 |
| District/ Field Office: | Moab |
| Resource Area: | Grand |
| Activity (program): | Oil & Gas |

**SECTION A. PROJECT INFORMATION**

| 1. Project Name: Well Site #136 | 4. Location Township: 275 | 5. Location Sketch: |
|---|---|---|
| 2. Key Observation Point: #15 on Hatch Pt. Rd. | Range: 21 E | The Knob Loop / Hatch Pt. Rd. / North / KOP / Well site |
| 3. VRM Class: Class II | Section: 24 | |

**SECTION B. CHARACTERISTIC LANDSCAPE DESCRIPTION**

| | 1. LAND/WATER | 2. VEGETATION | 3. STRUCTURES |
|---|---|---|---|
| FORM | Flat to rolling terrain | Simple forms created by vegetative patterns | ——— |
| LINE | Horizontal & diagonal | Weak & undulating | ——— |
| COLOR | Dark tans to orange | Light to dark green, mottled | ——— |
| TEXTURE | Smooth | Smooth to course | ——— |

**SECTION C. PROPOSED ACTIVITY DESCRIPTION**

| | 1. LAND/WATER | 2. VEGETATION | 3. STRUCTURES |
|---|---|---|---|
| FORM | Flat | Geometric & linear forms created by clearing | Cylindrical, geometric & angular |
| LINE | Horizontal (pad) Curved (road) | Strong irregular lines created by edge effect of clearing & rds. | Vertical, horizontal & angular |
| COLOR | Tan | Light green | Tan |
| TEXTURE | Fine to smooth | Patchy | Course |

**SECTION D. CONTRAST RATING __SHORT TERM __LONG TERM**

1. DEGREE OF CONTRAST / FEATURES

| | | LAND/WATER BODY (1) | | | | VEGETATION (2) | | | | STRUCTURES (3) | | | |
|---|---|---|---|---|---|---|---|---|---|---|---|---|---|
| | | STRONG | MODERATE | WEAK | NONE | STRONG | MODERATE | WEAK | NONE | STRONG | MODERATE | WEAK | NONE |
| ELEMENTS | FORM | | | ✓ | | | | ✓ | | | | ✓ | |
| | LINE | | ✓ | | | | | ✓ | | | | ✓ | |
| | COLOR | | | ✓ | | | ✓ | | | | | ✓ | |
| | TEXTURE | | | ✓ | | | ✓ | | | | | ✓ | |

2. Does project design meet visual resource management objectives? ☐ Yes ☑ No (Explain on reverses side)

3. Additional mitigating measures recommended ☑ Yes ☐ No (Explain on reverses side)

Evaluator's Names: R. Tumwater, R. Grimes, P. Jordan

Comments from item 2.
The Strong line created by the clearing for the road and the drill pad creates a contrast that will attract attention.

Additional Mitigating Measures (See item 3)
1. Relocate access road off from ridge.
2. Revegetate the edge of the drill pad with random clumps of trees and shrubs to break up the flat horizontal line.

**Figure 1.** BLM's visual contrast rating worksheet. Source: [13].

The actual "rating" is for the degree of contrast introduced by the activity-strong, moderate, weak, or none-with the form, line, color, and texture of the existing land/water, vegetation, and structures. Factors to be considered when determining the contrast rating

are distance, angle of observation, length of time the project is in view, relative size or scale, season of use, light conditions, recovery time, spatial relationships, and atmospheric conditions.

The BLM's implementation of visual contrast ratings does not include numeric scores, and there is no overall summation or determination of significance. The rating sheet shown in Figure 1 includes space for a narrative to describe whether or not the activity meets the VRM class objectives and recommendations for mitigation. The primary purpose is to diagnose whether mitigation is necessary and to provide design guidance if it is necessary.

### 3.1.3. BLM Contrast Rating's Distinctive Characteristics

The BLM's approach relies on previously determined visual resource objectives. The evaluation determines whether the visual change is compatible with the visual resource objectives; there is no absolute measurement of visual impact. This determination is guided by the contrast rating system, which is grounded in the Gestalt principles of perception and Modernist design principles. It deconstructs the view into its visual elements and spatial composition to determine the extent of visual contrast between a proposed project and its surroundings. Some of the distinctive characteristics of this approach are:

- The primary criterion is the project's visual contrast with the surroundings.
- The indicators are form, line, color, and texture contrast with the landscape features of land/water, vegetation, and structures. Other criteria may also be considered (e.g., season of use or atmospheric conditions), but their indicators are not described.
- The BLM inventories and evaluates the visual resources for all lands it manages and develops visual objectives for each management unit. The visual change is determined to be either compatible with the visual management objectives or not.
- It is a professional appraisal without public involvement.
- This analysis is limited to key observation points (KOPs), typically with visual simulations.

### 3.2. Berkeley Contrast Rating System
### 3.2.1. Background

In the late 1970s, the BLM and FS contracted with researchers at the University of California (UC) at Berkeley to use social science methods to evaluate the validity and reliability of visual attributes used in VIA and to recommend improvements based on their findings [14,15]. Nineteen photographs were selected of landscapes representing impact activities encountered by BLM. The "simulation" involved retouching the photographs to represent the scene's pre-impact condition. Slides were used to present these conditions to 54 students from UC Berkeley and UC Davis, 41 FS landscape architects, and 87 federal agency administrative employees. This remains one of the few major studies employing a landscape perception approach to investigate visual impact assessment methods.

A project's form, line, color, and texture contrasts with its surroundings were evaluated holistically for a given view, but also as associated with land/water, vegetation, and structures, as the BLM does in their contrast rating system. In addition, several other landscape descriptors that had appeared in the research literature were evaluated: ambiguity, compatibility, complexity, congruity, intactness, novelty, scenic beauty, unity, and vividness. Two additional dimensions were also evaluated: importance (of an activity with respect to scenic quality) and severity (of the visual impact). The testing involved measuring the landscape descriptors using the BLM's scale-none, weak, moderate or strong-but unlike the BLM qualitative assessment, these descriptors were assigned values from 0 to 3 [14]. Following these ratings, many of the participants were asked to list in rank order the factors that they thought most important in determining their scenic quality judgment and then again to list in rank order the factors they thought most important for assessing the severity of visual impact [15].

Feimer and Craik [14] (p. 6) found that "the average single-rater reliabilities for both direct and contrast was always less than 0.30 (where 0.70 or greater would be acceptable)".

They concluded that "the reliability of the BLM contrast ratings is only acceptable when independent ratings (i.e., not carried out collaboratively) of five or more raters are combined to form a composite index" [14] (p. 28). If the ratings are not combined into a composite index, 10 or more raters are required [16].

Smardon [15], (p. 56) considered the results of the rank order listing of which factors most influenced the scenic quality and visual impact judgments and concluded that Scale Dominance and Color Contrast were the most important factors, Form Contrast and Spatial Dominance were of medium importance, and Scale Contrast, Line Contrast, and Texture Contrast were the least important. Sheppard and Newman [17] used these results to prepare the *Prototype Visual Impact Assessment Manual,* which proposed a rating form to quantitatively evaluate the visual elements (form, line, color, and texture), scale dominance, and spatial dominance. These ratings are weighted and summed to obtain an index of visual impact severity. The range of possible index values is divided into five equal parts that are interpreted as negligible, weak, moderate, strong, or severe impact, though no evidence is presented to support this division. In addition, the Manual included text with extensive illustrative graphics to more completely define and explain the terms and provide examples of how to conduct the ratings. Smardon and Hunter [18] simplified this form, as shown in Figure 2, and it has been adopted for use by Maine's Department of Environmental Protection [19]. The sum of the ratings is an index of visual impact that is divided into four equal parts-weak/negligible, moderate, strong, and severe. These prototype procedures were not further evaluated.

### 3.2.2. Berkeley Contrast Rating's Distinctive Characteristics

The Berkeley approach grew out of agency-funded research in the late 1970s to evaluate the validity and reliability of observer-based visual impact assessment methods. Among the several visual factors evaluated, they found that the project's form, line, color, texture, and scale contrast, plus its spatial dominance with the surrounding landscape, provided the best explanation of perceived visual impact.

- The criterion is the project's visual contrast with the surrounding landscape.
- The indicators are form, line, color, texture, scale, and spatial dominance. It is assumed that contrast necessarily has a negative impact.
- The sum of the ratings becomes a numeric index with thresholds describing visual impact severity.
- It is a professional appraisal without public involvement.
- This analysis is limited to KOPs using visual simulations.

### 3.3. Guidelines for Landscape and Visual Impact Assessment

### 3.3.1. Background

Landscape character assessment (LCA) is a professional planning activity in the UK, comparable to mapping and assessing ecoregions or soil associations. Beginning in the 1990s, there was a concerted effort to inventory, describe, classify, and map the UK landscape following standard procedures [20]. This involved the use of existing maps and other data, but also extensive structured field surveys that included written descriptions, annotated maps and sketches, photographs, formal checklists, and descriptive rating scales. LCA forms a baseline description of the distinct patterns of qualities, attributes, and elements that characterize each landscape type. One type of landscape is not compared to another, though within a landscape type, society may assign value to a particular area to protect its character, for instance, through designation, or even to protect particular landscape elements, such as hedgerows or specific buildings. The LCA procedures have been revised every decade or so based on experience with their application, most recently in 2014 [20]. The landscape character of all areas in the UK has been classified and mapped and forms the basis for much of UK planning. This experience greatly influenced the European Landscape Convention, and now many countries are adopting and adapting it for their own use [6].

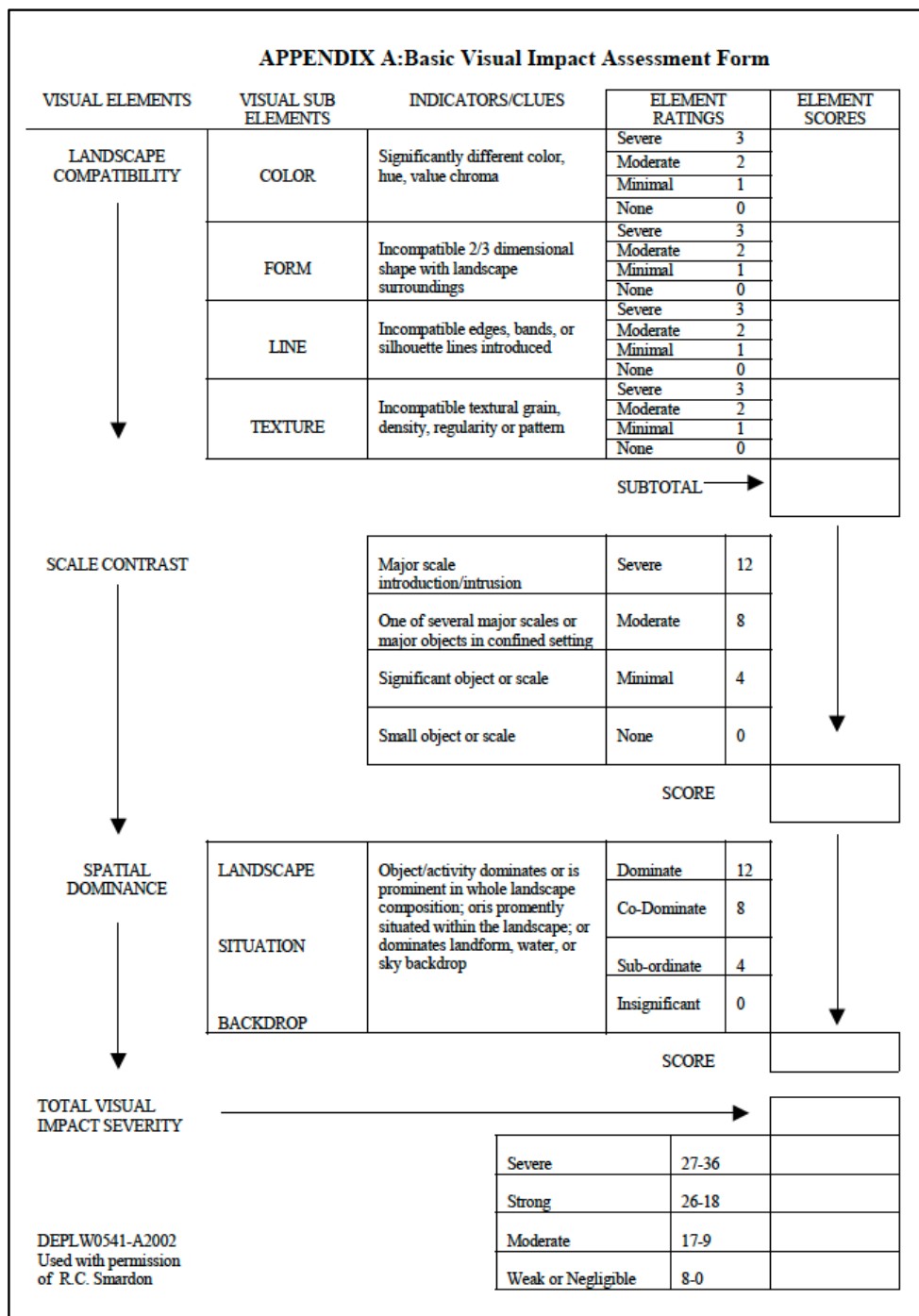

**Figure 2.** The contrast rating form used by the Maine Department of Environmental Protection. Source: [19].

Originally published in 1995, the *Guidelines for Landscape and Visual Impact Assessment* (GLVIA) is the recognized impact assessment method used in the UK and appears to have significant influence elsewhere, particularly in the Commonwealth of Nations. The Third Edition prepared by Carys Swanwick [21] reflects an evolution and refinement based on experience in its application and testing before the courts and related legal procedures. It is unique among the methods reviewed here in recognizing and separately assessing two distinct receptors of potential impacts-the landscape and people: "Landscape and Visual Impact Assessment (LVIA) is a tool used to identify and assess the significance of and the

effects of change resulting from development on both the landscape as an environmental resource in its own right and on people's views and visual amenity" [21] (p. 4).

### 3.3.2. General Procedure

The general outlines of the procedure to assess impacts on landscapes and people are the same for both receptors, as diagramed in Figure 3. The first step is to assess the sensitivity of the receptor, which is based on judgments of the receptor's susceptibility to change from the specific type of project being proposed, and the value associated with the receptor. An indicator of susceptibility might be based on a viewer's activity and the importance of scenery to that activity. An indicator of landscape value might be national, regional, or local designation.

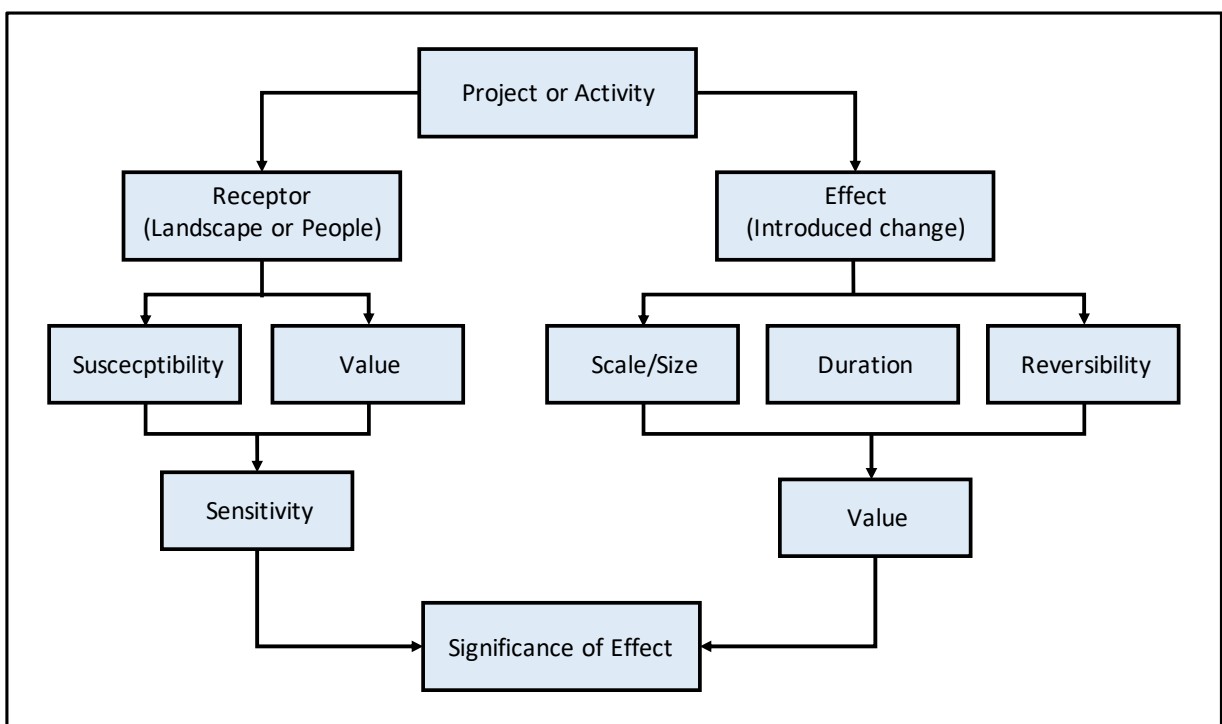

**Figure 3.** The GLVIA process for assessing the significance of effects. Adapted from [21].

The second step is to determine the magnitude of the change, which is based on the degree (i.e., size and scale), geographic extent, and duration and reversibility of the change. The degree might be changes to the composition or visual contrast in a view. Geographical extent might be the proportion of a landscape character area affected by a project. Duration might be described as short, medium, or long-term. Visibility analysis and visual simulations play an important role in determining magnitude.

Quantification of these effects is eschewed in preference for three to five descriptive ordinal categories that embody the assessment criteria [21]. However, exactly how to do this is left to the discretion of the LVIA analyst. Table 1 provides an example adapted from a quarry development and restoration GLVIA [22]. In addition, there is no prescribed procedure to combine the constituent assessments to describe susceptibility, magnitude, and overall impact significance. Some assessments profile all the criteria in a table, leaving the overall assessment to professional judgment, while others use a decision matrix to combine the criteria two at a time, as illustrated in Table 2. While the procedures described here are all based on professional judgment, the GLVIA does indicate that public consultation is an important part of the process. However, it is unclear how it is incorporated in a meaningful way.

**Table 1.** Methodology for Determining Visual Impact Magnitude and Receptor Sensitivity.

| Magnitude of Change | Evaluation | Receptor Sensitivity |
|---|---|---|
| Extensive or full intrusion to the existing view and/or the introduction of elements considered totally uncharacteristic in the view. Typically this would be where an application envelope would be seen in close proximity with a large proportion of the view affected with no/minimal screening or backgrounding and there would be a great scale of change from the present situation. | **High** | Elevated properties that are permanently occupied dwellings with prominent open views and occur within close proximity to the application envelope. Public Open Space, attractions, walking routes where surroundings are important to experience. Motorists and passengers on tourist routes. |
| Partial intrusion to the existing view and/or the introduction of prominent elements in the view. Typically this would be where an application envelope would be seen in views where a moderate proportion of the view is affected. There may be some screening or background which minimise the scale of change from the present situation. | **Moderate** | Elevated properties that are temporarily occupied dwellings with prominent open views. Motorists and passengers on arterial roads (with high numbers of daily vehicles). Commuter Rail Passengers. Users of urban shopping complexes. Users/workers of formal recreational landscapes (golf courses) |
| Low intrusion to the existing view and/or the introduction of features which may already be present in views. Typically this would be where an application envelope would be seen in distant views; where only a small proportion of the view if affected; where the effect is reduced due to a high degree of screening or background or where there is a low scale of change from the existing view. | **Low** | Temporarily occupied properties, be it elevated or otherwise, with restricted views. Users of sports specific playing grounds. Workers in their work place where setting not important to quality of working life. |

Source: [22] (p. 11).

**Table 2. Visual** Impact Significance Matrix.

| Sensitivity | Magnitude | | |
|---|---|---|---|
| | **High** | **Moderate** | **Low** |
| **High** | Major | Moderate to Major | Moderate |
| **Medium** | Moderate to Major | Moderate | Minor to Moderate |
| **Low** | Intermediate–Minor | Minor to Moderate | Minor |

Source: [22] (p. 11).

### 3.3.3. GLVIA's Distinctive Characteristics

The GLVIA is somewhat unique in that it is really two impact assessments-one for the landscape and another for people-that share a similar process. However, there are important differences, as well as similarities.

- The criteria are the receptor's (i.e., landscape and viewer) sensitivity (i.e., susceptibility and value) considered in combination with the magnitude of the project's effect (i.e., size/scale, duration, and reversibility).
- The identification and definition of indicators are not prescribed, though a set of indicators may come into common use over time for types of projects.
- A short narrative is used to describe ordinal levels for an indicator. It is left to the LVIA assessor to define these levels and then synthesize this information.

- It is a professional appraisal. The importance of public consultation is acknowledged but not provided for.
- Landscape impacts are mapped, sometimes with a GIS. Visual impacts are evaluated at selected viewpoints with visual simulations.

### 3.4. Queensland Scenic Amenity Methodology

3.4.1. Background

The Queensland (Australia) State Planning Policy for the coastal environment recognizes that "development may undermine the very scenic values that first attracted such development" [23]. The approach taken to implement this policy is most completely described in *What's in a view? SEQ Regional Scenic Amenity Study* [24], which has been summarized in a more recent guideline [23]. The Queensland scenic amenity method (SAM) is composed of two components, a baseline public scenic preference survey and the method of evaluating proposed scenic change.

3.4.2. Public Scenic Preference Survey

Several studies to determine the scenic amenity in South East Queensland were conducted for local areas prior to 2004. The diversity of approaches created challenges for presenting a consistent assessment procedure, and it was determined that a regional approach would be beneficial to regulators, developers, and the public.

A classification system was devised for landscape views. Four "visual domains" were identified to characterize the public's general notions of landscape types: bush, rural, urban, and coast. "Visual elements" are recognizable features that help determine scenic quality. These included natural elements, such as crops, pine forest, or bay water, and built elements, such as low residential buildings, fences, or roads. There were 110 elements identified by the field teams, which were simplified into 20 natural and 20 built visual elements.

A sampling scheme was developed by listing the visual elements that were particularly common or sometimes present within each visual domain. Field teams sought to obtain representative photographs to fill out this sampling scheme. If a photograph included a built element, then an attempt was made to shift the view to the right or left to take a photograph that did not include the built element but had similar natural elements. The sampling scheme also represented frequent combinations of visual domains, the distance from the photographer to visual elements, and whether the viewer's position was normal or elevated. The field teams were distributed throughout the region and obtained 2000 photographs using SLR cameras with a 55mm lens and 35 mm Kodak High Definition color film. These were winnowed to 440 photos that were relatively evenly distributed among the visual domain-by-visual element combinations. They were grouped into 22 sets of 20 photos, each printed at 5" × 7" that represented 5 levels of modification in each of the visual domains.

Members from the regional communities were recruited to compare the 20 photos in a set and rate their preference for the views on a scale of 10 for most to 1 for least preferred. The rating procedure made use of a grid with 10 columns; they were asked to place a photo under each column and could place up to 4 photos in a column.

After data cleaning and validation, there were valid ratings from 852 people that measured the scenic preference of 416 photos, which became the Scenic South East Queensland (SEQ) 2004 Image Library. Areas with a high scenic preference rating (SPR) are of state or regional importance and have a mean rating of 8 or higher; areas with medium SRP are of locally important scenic preference and have an SPR between 6 and 8. The details of this study are reported in [24].

The visual elements in each visual domain are outlined for each photo, as illustrated in Figure 4, and the proportion of the non-sky area is calculated. A regression analysis of the results is used to create a model that predicts mean SPR and 90% confidence intervals, the SEQ Scenic Preference Rating Assessment Tool (SPRAT) [25].

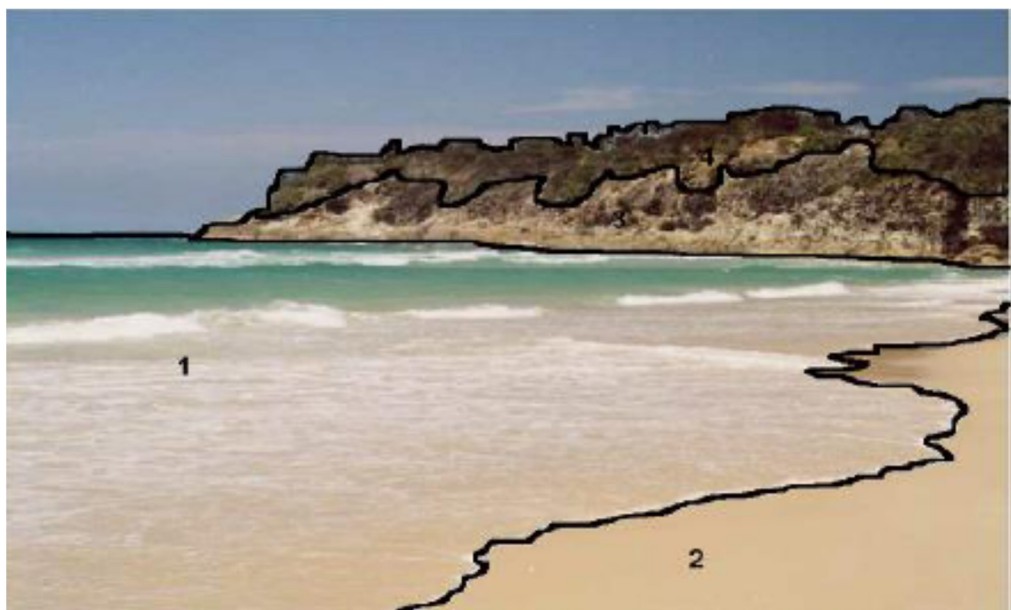

**Figure 4.** An example of a sketch delineating visual elements on a photograph. The area of each polygon is measured, and its proportion of the total non-sky area entered into SEQ SPRAT to estimate the view's scenic preference rating. In this image, 1 indicates bay and 2 sand beach. Source: [26] (p. 9).

Two methods are available to determine the scenic preference of new locations using a minimum of three photographs taken to represent views from the development site [26].

- Search the Scenic SEQ 2004 Image Library for three photos that most closely match each photo representing the location of interest and calculate their average scenic preference.
- Measure the proportional area of visual domains and visual elements that make up a view in a photograph, and use the SEQ's SPRAT spreadsheet to predict scenic preference.

### 3.4.3. Scenic Amenity Methodology (SAM)

The above procedures have been extended to become a VIA method for development within the coastal zone [26]. When a development is proposed, a minimum of three locations are photographed to represent the pre-change SPR. These viewing locations must be "the most highly used and affected public viewing location within a maximum distance of five kilometers" [26] (p. 5). The visual domains (e.g., coast, bush, rural, and urban) and elements (e.g., low residential building, road, pine forest, crops) are described for each photo, and the SPRAT spreadsheet is used to calculate the SPR, as described in the previous section.

The three photographs are used to prepare photo-realistic simulations. The visual domains and elements are measured for each simulation, and the SPR is determined using SPRAT. SPRAT is also used to compare the SPR for the existing and simulated conditions and determine if the difference is statistically significant. If the SPR for at least two-thirds of the photo-simulation pairs is significantly different, they are evaluated according to the acceptable level of change shown in Table 3. Proposals with an unacceptable level of change must be redesigned by modifying the location of project structure elements, reducing their volume, or screening them with vegetation. The revised design is re-evaluated. If the level of change is still unacceptable, then the project should not be approved.

**Table 3.** Acceptable level of change.

| | Pre-Change SPR | Lowest Acceptable Post-Change SPR |
|---|---|---|
| | 10.0 | 10.0 |
| **Areas of high scenic preference** | 9.0–9.9 | 9.0 |
| | 8.0–8.9 | 8.0 |
| **Areas of medium scenic preference** | 7.0–7.9 | 7.0 |
| | 6.0–6.9 | 6.0 |

Source: [27] (p. 10).

3.4.4. SAM's Distinctive Characteristics

SAM is the most objective of the KOP methods reviewed here, in the sense that there is little opportunity for assessor bias when comparing the pre- and post-construction simulations. Nonetheless, its results are dependent in part on the specific view photographed by the assessor-a limitation of all the KOP methods. Its distinctive characteristics include:

- The criterion is predicted change in scenic preference rating.
- The indicators are the area of visual domains and elements measured from an existing photograph and a photo simulation of the project.
- A public preference survey established a predictive equation based on the indicators. The equation is applied to pre- and post-development views. If the difference is not within acceptable levels of change, then the project must be revised, or it is denied.
- The evaluation mode is based on a statistical analysis of a large public preference survey. Professionals implement this approach, including the selection and measurement of views to evaluate.
- This analysis is limited to three viewpoints using visual simulations.

*3.5. The Spanish Method*

3.5.1. Background

As a member of the EU, Spain is responsible for assessing the landscape and visual impact associated with large projects. Many VIAs include a map showing areas from which a project may be visible-the viewshed-but the actual assessment is based on photo simulations from a limited number of viewpoints. Hurtado, Fernández, Parrondo, and Blanco [27] identified the need for an unambiguous method to evaluate the visual impact of wind farms on the surrounding landscape, but in particular nearby villages. They proposed using GIS software to calculate five original coefficients to describe the potential visual impact of wind farms being proposed in Spain.

Manchado, Gomez-Jauregui, and Otero [28] reviewed the "validity, efficiency and limitations" of the original Spanish Method. They implemented revisions to the definitions and methods to improve calculation efficiency. Each of the indices is standardized to have a value between 0.0 and 1.0. Though it was developed to evaluate wind farms, it is applicable to any highly visible structure-transmission towers, communication towers, power plant cooling towers, or even large buildings.

The Spanish Method requires a visibility analysis conducted using raster data, with an indication of the number of wind turbines visible from each cell. The visibility analysis may include the screening effect of vegetation and structures or not. Several of the coefficients described below investigate the impact on a settled area (e.g., a "village") but also could be applied to any area of concern, such as a scenic resource (e.g., a lake) or the whole study area. The following summarizes how the five coefficients are calculated and their interpretation.

3.5.2. Spanish Model Coefficients

**Visibility coefficient *a*: General visibility**. The visibility results for a village are extracted from the viewshed. The number of cells within this area is denoted by *n*. The number of turbines visible in cell *i* is *Xi*, and *WT* is the total number of wind turbines.

$$a = \frac{\sum_{i=1}^{n} \frac{Xi}{WT}}{n}$$

The interpretation coefficient *a* is the average number of wind turbines seen from the village or the probability of seeing a wind turbine within the village [28]. A modification to this coefficient only considers areas of visibility within the village. The ratio of cells with visibility to the total number of cells is suggested as a simpler alternative form of coefficient *a*.

**Coefficient *b*: Built-up area with visibility**. This coefficient requires a comparison of the viewshed to the location of residences or buildings within a village. This might be obtained from emergency address location data or created in a GIS from aerial imagery. The number of buildings within a village with some visibility of the project is *VB*, and the total number of buildings is *TB*.

$$b = \frac{VB}{TB}$$

The interpretation of coefficient *b* is the percent of buildings in a village with the visibility of a wind turbine [28]. A modification to this coefficient is suggested that adjusts the viewer's eye level to be from the upper stories of a building, though this would require additional information about the buildings. Alternately, some buildings might be considered more sensitive than others, such as designated historic public buildings.

In the absence of information about buildings, it would be possible to set a population density threshold above which a cell is considered to represent a building; then *TB* would be the number of such cells, and *VB* would be the number with visibility.

**Coefficient *c*: Relative position**. The visual effect of a row of structures, such as wind turbines along a ridge or transmission structures, is greatest when viewed perpendicular to the row and least when viewed along the row, where individual structures overlap each other. The coefficient is the product of factor describing ranges for the number of turbines (*m*) and the relation of the viewer to the line of structures (*v*).

$$c = v * m$$

The values of *m* and *v* have not kept pace with changes in the size of wind turbines and projects; the general approach presented also has been shown to have significant problems [28].

**Coefficient *d*: Distance**. Distance (*x*) from the cell to the nearest turbine is measured. For distances less than 500 m (0.31 mi), the value of *d* is 1.00, and for distances greater than 6 km (3.73 mi), *d* is 0.10. For distances between 0.5 and 6.0 km:

$$d = 1.05 - 0.0002 * x$$

Manchado et al. [28] recognize that the relation of distance to the degree of visual intrusion may not be linear. In addition, the values of *d* may need revision to reflect the increasing size of wind turbines.

**Coefficient *e*: Population**. The Spanish Method was originally designed for rural areas where villages often had very small populations. Coefficient *e* ranged from 0.20 for villages with a population of 1 to 5 people to 1.00 for villages where the population was 300 or more people. Manchado et al. [28] indicated that coefficient *e* is prone to controversy and recommend that issues associated with population be handled through an effective program of public engagement.

**Global assessment**. In their original proposal, Hurtado et al. [27] proposed a partial assessment index (PA1) as the product of coefficients *a, b, c,* and *d,* and PA2 added coefficient *e*. However, after a decade of experience with the Spanish Method, Manchado et al. [28] do not recommend the use of these global indicators:

> "Our opinion is that the Spanish Method is rich and valuable when coefficients a–e are individually analyzed, but we do not trust too much on the meaning of the integration of these coefficients into a single number. Among many other reasons, such a general coefficient could compensate possible extreme values of their individual coefficients, what not always helps to make a good analysis. Moreover, the Spanish Method can be a good tool to understand and to express the visual effects, but the final assessment must be conducted by means of participation and agreement in some manner" [28] (p. 764).

Instead, they seem to be recommending that the VIA professionals treat coefficients *a–e* as diagnostic tools. They might point out that a doctor considers the results of each medical test when advising us about our health and would not simply combine the results into a single index. However, any critical value might be moderated by the other values in the index. They seem to be recommending that the interpretation of these coefficients be conducted in a public forum to achieve a common agreement. Manchado and her colleagues have continued to improve the Spanish Method's coefficients and, after several iterations, made it available as the online tool MOYSES 4.0. However, as the size of turbines and projects has grown, the required memory and time for the calculations have greatly expanded beyond the capacity of an online tool. The most recent application assessed the visual cost of energy facilities in regional areas, optimized the coefficients for parallel computing, and used a supercomputer to perform the demanding calculations [29]. This expanded capacity allows calculating the indicators for each and every point of the study area.

### 3.5.3. Spanish Method's Distinctive Characteristics

The SP2 criteria and indicators are well-grounded in landscape assessment and perception research. The result is a GIS-based procedure that is objective and relatively independent of potential assessor bias. Its major limitation is that its validity has not been rigorously evaluated.

- The criterion appears to be the project's visibility as experienced by viewers in the surrounding landscape.
- There are five coefficients that measure general visibility, visibility from built-up areas, the project's relative position or orientation to the viewer, the distance between the viewer and project, and the size of the viewer population.
- The coefficients are interpreted by professionals as diagnostic tools rather than summed into a single index with rigid thresholds.
- Professionals implement this approach.
- The analysis is done with a GIS.

### *3.6. Maine's Wind Energy Act*
### 3.6.1. Background

In 2008, Maine passed the Wind Energy Act (WEA), which declared an emergency for renewable energy development to address energy independence and reduce greenhouse gasses. It also recognized that there was heightened controversy and a lack of clarity around the evaluation of visual impacts. In response, it mandated a new approach to assessing the visual impacts of wind energy development. The WEA recognizes that being "a highly visible feature in the landscape is not a solely sufficient basis for determination that [a project] has an unreasonable adverse effect". Among the WEA's provisions, the consideration of visual impacts was limited to "the scenic character or existing uses related to scenic character of the scenic resource of state or national significance" (SRSNS). Eight

categories of SRSNS were identified, and in most cases, they were embodied in a database that included an evaluation of scenic value or significance. In addition, the WEA established seven criteria to be used to evaluate the scenic impacts.

At about this time, Tveit, Ode, and Fry [30] published an influential review of the visual landscape character literature that proposed a hierarchical framework where an abstract concept, such as visual impact, is deconstructed into several abstract dimensions or criteria. One or more physical characteristics are identified to measure a specific aspect of a criterion; these indicators may be physically measured (e.g., area, visitor counts) or with rating scales (e.g., beliefs about future behavior). Finally, the responsible agency requires recommended thresholds to determine whether an indicator value represents a positive or no visual impact, an adverse visual impact, or an unreasonable adverse visual impact that is cause for denying a project permit application. The professionals preparing wind project VIAs in Maine, in conjunction with a peer-review process, have evolved the implementation of this framework.

The WEA criteria may be divided into two categories based on users and SRSNS. While the implementation varies slightly from project to project, the goal is to have at least two indicators describing each criterion, and these indicators may be qualitative or quantitative. General thresholds were defined to interpret the importance of an indicator's value as none, low, medium, or high. A combination matrix is used to determine the indicators into a single five-point rating for a criterion: Low, Low-Medium, Medium, Medium–High, and High; areas without visibility have a rating of None. Further detail is available as described by Palmer [31]; the review of the Bingham Wind Project demonstrates how these guidelines are applied in a VIA [32].

### 3.6.2. WEA Impact Criteria

The WEA seeks to protect designated SRSNS and the users of those SRSNS for whom their significance or scenic value is important. The following description of the seven criteria provides examples of indicators that have been used.

Where user criteria have the potential to be an important factor, Maine has encouraged the developer to conduct a survey of users at the location of one or more visual simulations. The criteria oriented toward users are:

C. The **expectations** of the typical viewer.

- The intercept survey includes a question concerning expectations while at the SRSNS and then whether the change in view affects those expectations.
- Expectations can be inferred from an SRSNS's classification of the recreation opportunity spectrum [33–35].

E.1. The **extent, nature, and duration** of potentially affected public uses of the scenic resource of state or national significance.

- It is the nature of some activities that scenery plays a more important role in their enjoyment than in other activities. This is often based on the VIA professional's judgment, though some guidance is provided by recent research based on visitor surveys [36].
- The extent and duration of use can be observed during the intercept survey; other times, it is inferred from field observations. Whether the use is high, medium, or low is often based on the VIA professional's judgment, though recreation guidelines provide some guidance [33,34].

E.2. The potential effect of the generating facilities' presence on the public's **continued use and enjoyment** of the scenic resource of state or national significance.

- The intercept survey includes a question concerning how the project will affect the continued use of the SRSNS.
- The intercept survey includes a question concerning how the project would affect enjoyment while at the SRSNS.

The five criteria oriented toward SRSNS are:
A. The **significance** of the potentially affected SRSNS.

- The level of significance is frequently identified in the relevant SRSNS database. For instance, the National Register of Historic Places indicates whether the site has national, state, or local significance; the inventory of lakes and rivers evaluates scenic value as outstanding, significant, or less.

B. The **existing character** of the surrounding area.

- Maine's public and government believe it is important to protect the remote character of its forest lands. Standard procedures are used to identify "remoteness" [33–35].
- An investigation of an area's "image" or recognition as indicated by its popularity as a tourist destination through an internet search, as well as the tourism literature more generally [37].

D. The expedited wind energy development's purpose and the **context of the proposed activity**.

- Projects receive a higher rating by limiting the need for new associated facilities, such as transmission lines and roads.
- The greatest scenic impact occurs with the introduction of the first project; successive projects of the same size each have a lower incremental impact [38]. Clustering projects in an area in order to leave other, higher-quality areas unaffected is desirable.

F. The **scope and scale** of the potential effect of views of the generating facilities on the SRSNS, including but not limited to issues related to the **number and extent of turbines** visible from the SRSNS, the **distance** from the SRSNS, and the **effect of prominent features** of the development on the landscape.

- Create a table that shows the percent of an SRSNS's area with hub visibility of at least 0, 1, 2, . . . n turbines. A table is proposed to translate the number of turbines visible in the tenth percentile into a rating.
- Using the same table, what percentage of the SRSNS's area has visibility of at least one turbine hub? The rating is low if the area is less than a third and high if it is greater than two-thirds.
- There is a strong relationship between visual impact and the distance to the closest turbine for which there is substantial visibility (e.g., seeing the turbine hub, not just a blade tip). The WEA establishes 3 miles as high impact, 3 to 8 miles as medium, and beyond 8 miles as low impact. Turbines have substantially increased in size, and these distance thresholds should be re-evaluated.
- Identify prominent landscape features seen from the SRSNS. When looking at the landscape feature, if portions of the project are visible within a 30° horizontal arc, then the rating is high; if the project is beyond 120°, it is low.

The results of the ratings are generally arrayed in a table with the criteria across the top, and the SRSNS listed down the side, as illustrated in Table 4. The ratings are treated as ordinal, and there is no attempt to calculate a numerical average. There is some flexibility in determining whether this array of ratings constitutes an unreasonable scenic impact. If there is a criterion with a rating of high, that may be sufficient grounds to consider the scenic impact unreasonable, particularly if the area of the SRSNS is large. If a substantial proportion of the area within 8 miles of the project-say 10%-includes SRSNS with ratings of medium or higher, that might constitute an unreasonable scenic impact. Similarly, if most of the SRSNS in the study area have ratings of medium or higher, then that might constitute an unreasonable scenic impact. Recent guidance is to determine a project's impact as unreasonable if the significance is high or medium as seen from a high-value scenic resource, or if the significance is high from one or medium from several medium-value scenic resources.

**Table 4.** Evaluation Criteria Rating Summary.

| Scenic Resources of State or National Significance | Scenic Impact Evaluation Criteria | | | | | | | Overall Scenic Impact |
|---|---|---|---|---|---|---|---|---|
| | **A** | **B** | **C** | **D** | **E.1** | **E.2** | **F** | |
| **Historic Sites** | | | | | | | | |
| Arnold Trail to Quebec | Low-Med | Medium | Medium | Med-Low | Med-Low | Medium | Low | Low+ |
| Bingham Free Meetinghouse | Low | Medium | Low | Med-Low | Low | None | None | None |
| **National or State Park/Designated Pedestrian Trail** | | | | | | | | |
| Appalachian National Scenic Trail | High | Medium | High | Med-Low | Medium | None | None | None |
| **Great Ponds** | | | | | | | | |
| Bald Mountain Pond | Medium | Medium | High | Med-Low | Low | Med–High | Low | Low+ |
| Jackson Pond | Medium | Medium | Medium | Med-Low | Low | None | None | None |
| **Segment of a Scenic River** | | | | | | | | |
| Wyman Lake | Medium | Medium | Med-High | Med-Low | Low | Medium | Med–Low | Low+ |
| Kennebec River: Augusta to the Forks | Medium | Medium | Med-High | Med-Low | Low | Medium | Low | Low+ |
| **Scenic Turnout on a Scenic Highway** | | | | | | | | |
| Old Canada Road Scenic Byway Turnout | Low-Med | Medium | Medium | Med-Low | Low | None | None | None |

Notes: The Evaluation Criteria are: (A) Significance of resource, (B) Character of surrounding area, (C) Typical viewer expectation, (D) Development's purpose and context, (E.1) Extent, nature, and duration of uses, (E.2) Effect on continued use and enjoyment, and (F) Scope and scale of project views. Source: [32].

3.6.3. Maine WEA's Distinctive Characteristics

This approach has been developed organically by VIA professionals responding to legislated VIA criteria.

- The criteria are the value of designated scenic resources and the significance of the visual impact.
- The indicators of the value of the scenic resource are the significance of the resource, surrounding character, the context of the site, and the scope and scale of the project. The indicators of the impact's significance are user expectations, extent, nature and duration of use, and effect on continued use and enjoyment.
- The professional assessor determines how to measure the indicators as having a low, high, or medium effect and combines them to describe the value of the scenic resource and the significance of the visual impact.
- Indicators concerning users are measured using an intercept survey at simulation viewpoints. Professionals measure indicators concerning the significance of the impacts.
- The analysis uses both a GIS analysis and visual simulations at KOPs.

## 4. Discussion and Conclusions

### 4.1. Comparative Summary

The purpose of this article is to acquaint the reader with a sample of the diverse methods used to evaluate visual impacts. At the beginning of the article, seven factors to consider were posed and commented on at the end of each method's presentation. Table 5 pulls those threads together in a comparative summary. The table first lists the basic themes used in evaluative research to describe a program or process: objectives, criteria, indicators, and standards or thresholds. It then compares three attributes that describe how the method

is implemented: quantitative or qualitative measurement, professional assessors or public perception, and a GIS or KOP framework.

The **objective** of each method is to assess visual impacts, but there are slight differences among them. The BLM's method is part of a comprehensive approach to visual resource management. While several approaches imply that impacts on both the landscape and viewers need to be considered, only GLVIA explicitly recognizes them as distinct receptors and evaluates them separately.

There are larger differences in the **criteria** or conditions that must be met in order to achieve the objective. The BLM and UCB focus on the visual contrast between the existing landscape and the introduced development, on the assumption that it is generally desirable that a project's or management activity's visual presence be minimized. GLVIA and WEA focus on the significance or sensitivity of the receptor and the magnitude of the visual change. Alternatively, SAM applies an empirically derived formula to calculate the scenic value of an existing view and the same view with the proposed development; scenic impact is the difference between the two values.

Because the criteria are quite different, the measurable **indicators** used to assess each criterion are also very different. Form, line, color, and texture contrast are the primary indicators used by the BLM and UCB. SAM measures the area occupied by visual elements (e.g., types of land cover or features) in a photograph. GLVIA and WEA assign an ordinal rating based on a narrative characterization of each level. SP2 calculates five coefficients that measure the degree of visual exposure and the size of the viewer group. Sometimes indicators are considered independently (e.g., SP2), and sometimes they are combined into a single index (UCB).

If the purpose of a VIA is the determination of the visual impact's significance or seriousness, then what **standard** or threshold is used to separate the acceptable from the unreasonably adverse impacts? The BLM uses visual management objectives to describe the standard for different areas. UCB and SAM have established numeric thresholds to distinguish the intensities of visual impact. The other methods allow for significant professional interpretation of the indicators rather than prescribe a standard.

A major distinction among these methods is whether the measurement is **quantitative** or **qualitative**. Methods can differ even when the same indicators are used-BLM uses qualitative description, and UCB applies numeric rating scales that are summed to form an visual impact index. SAM and SP2 are also very quantitative, while GLVIA and WEA tend to use ordinal quantification with narrative descriptions.

Another major distinction is whether the **public** is involved in the assessment or is it only based on a **professional** appraisal. In general, these methods are driven by professionals. However, the original SAM database and analysis were based on a large public preference survey of coastal landscapes. The WEA frequently includes an intercept survey of the public found at or near the simulation viewpoints.

A third major distinction is whether the method is primarily based on evaluating the change as seen from a few selected **KOP**s or uses a **GIS** to model the visual change (i.e., more than a basic viewshed) throughout the whole study area. BLM, UCB, and SAM are KOP-based, SP2 is GIS-based, and GLVIA and WEA incorporate both approaches.

Finally, Table 5 lists at least one **unique aspect** of each method that sets it apart from the other methods. The BLM's approach to contrast rating was designed so that non-design professionals could apply it, and they have provided training in their approach for over 30 years. The BLM also expects the assessment to be conducted by a team of independent assessors in the field. A research group at UCB was commissioned to evaluate the BLM and other visual impact indicators. One result was the quantitative approach to evaluating contrasts. GLVIA is particularly important for its recognition that visual impacts affect two very different receptors-landscape and viewers-and that they should be assessed independently. SAM is the only example to employ a rigorous analysis of public scenic preferences-852 people evaluated 416 views. SP2 is the only method based solely on a GIS

analysis. WEA is the only method that surveys public perceptions of the proposed project from the simulation viewpoint.

**Table 5.** Summary of Six Approaches to Visual Impact Assessment.

| Attributes of Approach | Visual Impact Assessment Method | | |
| --- | --- | --- | --- |
| | **BLM** | **UCB** | **GLVIA** |
| **Objective** | Analyze potential visual impacts of projects and management activities. | Assess potential impacts on existing scenic resources and aesthetic uses | Assess effects and significance of visual change on receptors. |
| **Criteria** | Contrast | Contrast | Receptor (landscape and viewer) sensitivity and magnitude of effect. |
| **Indicators** | Form, line, color, texture | Form, line, color, texture, scale, spatial dominance | Sensitivity: susceptibility and value. Magnitude: size/scale, duration, reversibility. |
| **Standard** | Visual management objective | Numeric rating threshold | Interpretation by VIA assessor. |
| **Quantitative-Qualitative** | Narrative description | Quantitative rating scales | Narrative description with ordinal rating. |
| **Professional-Public** | Professional | Professional | Professional |
| **GIS-KOP** | KOP | KOP | GIS and KOP |
| **Unique aspect** | Intended for trained non-design professionals. Evaluated in the field by a team of independent assessors | Evolved from research evaluating BLM reliability and validity | Recognizes impacts to landscape as separate from impacts to viewers. |
| Attributes of Approach | Visual Impact Assessment Method | | |
| | **SAM** | **SP2** | **WEA** |
| **Objective** | Preserve the scenic amenity of the coast in a manner that maintains its natural character | Provide an objective characterization of the visual impact of a wind farm | Keep wind development from unreasonably affecting scenic resources or uses related to scenic character. |
| **Criteria** | Scenic preference rating | Visibility experienced by viewers | Value of scenic resource. Significance or size of impact. |
| **Indicators** | Area on a photo of visual domains/elements | Five spatial coefficients | Resource significance, surrounding character, site context, project scope/scale. User expectations, extent, nature and duration of use, continued use and enjoyment. |
| **Determination** | Numeric index thresholds | Interpretation by VIA assessor | Guidance from regulations |
| **Quantitative-Qualitative** | Quantitative regression-based index | Quantitative coefficients | Ordinal rating |
| **Professional-Public** | Based on public survey; professional implementation | Professional | Professional; survey of public at simulation viewpoints. |
| **GIS-KOP** | KOP | GIS | GIS & KOP |
| **Unique aspect** | Method derived from large public survey | Requires extensive computing resources; exploring use of parallel computers | Regularly conducts surveys from simulation viewpoint. |

Note: The visual impact assessment methods are: BLM Contrast Rating System (BLM), Berkeley Contrast Rating (UCB), Guidelines for Landscape and Visual Impact Assessment (GLVIA), Queensland Scenic Amenity Methodology (SAM), The Spanish Method (SP2), Maine Wind Energy Act (WEA).

*4.2. Recommendations*

A recent review of the peer-reviewed literature in the domain of landscape assessment and perception found relatively little activity in the area of visual impact assessment [39]. This is important because it is not clear how relevant the landscape assessment and percep-

tion literature is to visual impact assessment. Visual impact assessment is fundamentally about the differences between two similar scenes, and landscape assessment and perception are fundamentally about classifying different landscapes of comparing different views. Therefore, this review concludes by making recommendations for visual impact assessment research and best practices.

### 4.2.1. VIA Gold Standard

The first need is to establish a recognized gold standard for determining visual impacts. It is proposed that this requires surveying the people who use the location that will be impacted while they are engaged in the activities that they normally do there. It will be necessary to understand their experience and attitudes before and after the project is built. Palmer [40] has shown how to use effect size from these survey responses to determine when the impact is unreasonable. A common protocol for such case studies needs to be developed, perhaps by the Visual Resource Stewardship group.

It may be possible to implement most of the gold standard protocol while conducting a VIA and then completing a post-construction audit to evaluate the VIA's findings. In other cases, researchers may independently implement the protocol when a controversial project is proposed. These case study data should be made publicly available, including high-resolution photo simulations, survey responses, and GIS layers. In some cases, it may be possible to "test" an existing VIA method against a gold standard case study. Academics could explore new GIS or KOP approaches to VIA by using the open data case studies.

It is accepted, even expected, that an environmental impact assessment includes field inventories describing potential natural receptors, such as birds and bats, and it is a frequent condition in the development permit that these receptors be monitored for a period after construction. A similar evidence-based approach should be applied to the effects on people. The monitoring results should be compared to the VIA predictions. As this protocol is implemented across multiple projects, a meta-analysis of results for each method should be prepared. The results of this research could be invaluable for setting standards or thresholds.

### 4.2.2. Multi-Method VIAs

In addition to establishing a gold standard, VIA practice would be strengthened by employing more than one method. This type of triangulation is used in research as a way to validate (or not) a study's results and to improve the methods. This would be particularly effective if one method was GIS-based and the other KOP-based. The VIA would compare the results of the GIS analysis with the KOP locations e.g., [41,42]. In some situations, KOP analysis can be used to calibrate the GIS results measuring the seriousness of the impact on the whole study area [43].

### 4.2.3. Test the Validity of VIA Methods

There has been relatively little research testing the validity of the VIA inputs, such as photo simulations, levels of viewer sensitivity, or visual impact criteria.

**Photo simulations**. The general acceptance of photo simulations as valid representations of future visual conditions is based on landscape perception research that compared the correlation of scenic ratings of photographs and views in the field. Correlation is the wrong measure of whether a change in a view is significant-Cohen's d is more appropriate [40,44]. There is recent research that indicates that photographs systematically underestimate visual impacts as judged in the field [45,46]. In addition, we need to determine the most valid way to present simulations. It is known that both the photograph's horizontal angle of view (i.e., lens focal length) and the size of the image affect the judgment of photo simulations [45,47].

**Viewer sensitivity**. It is reasonable to consider visual impact as involving the viewer's sensitivity combined with the magnitude or significance of the visual change. In general, the basis for viewer sensitivity in VIA methods is the opinion of professionals who have

little or no expertise in how sensitivity varies among viewer populations and their activities in the landscape being assessed. As an example of the lack of agreement, the GLVIA considers residents' views as among the most sensitive, but the WEA excludes them from consideration. Palmer and English [35] evaluated the scenic sensitivity of over 100,000 visitors engaged in 24 recreation activities in U.S. National Forests; similar investigations need to be conducted for additional activities and settings.

Some of the above VIA methods use the number of viewers and the duration of the view as indicators of viewer sensitivity. However, these seem poorly considered. For instance, what is the equivalence between 8300 daily viewers on the White Mountain National Scenic Byway (which is also Interstate Highway 93) and a few hundred hikers on the Appalachian National Scenic Trail, both with equivalent views of the project? Recreation managers and researchers have been working on the appropriate intensity of use in different settings for decades [33], but it has yet to become a priority for landscape perception researchers. Or another example, suppose someone is driving towards a large wind energy development in the wide-open landscape of the western U.S., where they may watch it grow in visual magnitude for 20 or 30 miles. Compare this to the resident who lives in a well-wooded neighborhood in New England and, when driving to or from home, passes through a transmission line right-of-way in seconds, though it may be several times every day. What duration would be considered "high" and why? Maybe the whole concept of duration as an indicator is misplaced. Researchers have found that an accurate aesthetic response is formed after just a glance at a scene [48–50]. Some research has found that after much longer exposure and familiarity, many people find wind turbines or power lines less offensive [51]. VIA professionals should stop guessing and conduct the research necessary to understand the validity of these indicators.

**Criteria and indicators**. VIA criteria and indicators should also be subjected to rigorous analysis of their validity. In general, this involves conducting the VIA and comparing the results to a gold standard audit of the built project. The BLM contrast ratings are based on a Modernist theory of design [11,52]. The evaluation of VIA indicators being explored at the time found mediocre validity when compared to scenic beauty change scores [53]. While the SAM method is based on statistically significant regression analysis to explain the scenic preference of coastal landscape views, there was no evaluation of the method's validity when applied to VIA. The other methods' indicators have similarly not been subjected to rigorous validation as they are applied in VIA.

### 4.2.4. Test the Reliability of VIA Tools

Reliability is the evaluation of whether different professionals obtain the same result when presented with the same information. Examples of where reliability may be a concern include the accuracy of photo simulations, agreement among independent assessors' evaluations, and the mapping of project visibility.

**Photo simulations**. There has been considerable discussion in the UK about the validity and reliability of photo simulations. This led The Landscape Institute to revise its recommendations for simulation photography, preparation, and presentation [54]. Recent guidance from the U.S. National Park Service describes a number of inaccuracies found in professionally prepared photo simulations and provides guidance on how to detect them [55]. More attention needs to be given to the accuracy and reliability of photo simulations and other media used to represent visual change.

**Assessors**. There is some research on the reliability of the numerical rating scales used for VIA, and it recommends that 5 to 10 or more independent assessors be used to achieve a professional level of reliability [16,56]. No reliability studies have been conducted for indicators that use a qualitative measurement. This is particularly important since even large wind energy projects use no more than three or four assessors, well under the number necessary to be confident in the reliability of the results. Research also needs to be conducted on how training affects reliability, possibly leading to a training standard for professionals using particular VIA methods.

**Visibility mapping**. Most VIAs include a map of a project's potential viewshed, though it seldom plays a meaningful role in evaluating the visual impacts unless the method is based on further GIS modeling. Past research has found that viewsheds have a modest level of reliability or validity. That is, many areas that are in the GIS-determined viewshed turn out not to have visibility of the project, and conversely, many areas with visibility are not in the GIS-determined viewshed [57,58]. However, the publicly available sources of terrain and land cover elevation data are rapidly evolving, and their reliability and validity need to be evaluated [59,60]. In addition, attention should be given to whether the parameters of the analysis are meaningful and whether the results are understood as intended [61].

### 4.3. Conclusions

This article has reviewed several diverse methods that are currently being used to assess visual impacts. Diversity can be a healthy sign, but it is important not to simply embrace the newest suggestions for conducting VIA without demonstrating their efficacy. To do this, it is necessary to establish a common gold standard that is acceptable to the public, permitting agencies, developers, and those conducting VIAs. It should become common practice, at least for large projects, to include pre- and post-construction measurements of the gold standard in order to validate and improve VIA methods. In addition, more consideration should be given to how to demonstrate the reliability of the methods used to prepare a VIA.

Visual impacts are often among the public's primary concerns when a large development project is proposed. VIAs are the primary document used in the permitting process to determine whether the visual impacts are somewhat or unreasonably adverse. It is the responsibility of the professionals engaged in this field to assure that their methods are valid and reliable. This can be achieved by bringing an empirical, scientific approach to bear on the conduct of VIAs.

**Funding:** This project has been funded wholly or in part by the United States Environmental Protection Agency under assistance agreement CB93641401 to the Chesapeake Bay Trust. The contents of this document do not necessarily reflect the views and policies of the EPA, nor does the EPA endorse trade names or recommend the use of commercial products mentioned in this document.

**Institutional Review Board Statement:** Not applicable.

**Informed Consent Statement:** Not applicable.

**Data Availability Statement:** Not applicable.

**Conflicts of Interest:** The authors declare no conflict of interest.

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
