# Peer review of "A Diversity of Approaches to Visual Impact Assessment"

_land, doi:10.3390/land11071006_

Round 1

Reviewer 1 Report

All the comments and suggestions are included in the atatchment.

Author Response

See attached PDF

Reviewer 2 Report

The article is engaging and rich in informative content about the Visual Impact Assessment method's differences, similarities, and history. The author's research perspective and purposes are evident since he provides recommendations to improve the reliability of VIA tools and techniques according to his long experience in the field. All the article's Sections are clear, well-structured, and understandable also by non-expert people. The standardised criteria used by the author to compare the six shown methods make understanding the text simple, and the final recommendations support the starting statements and the author's intentions.

Whether the article can be published in its current form, I can provide that author with two minor revisions:

1) In table 5, the item "Unique aspect" inside the "attributes of approach" field does not match its repetition (in the last raw table, indeed, it is referred to as "Other comments"). Since the table aims to remark on each approach's uniqueness, I suggest replacing the term.

2) In Section 4.2 (Recommendations), the author cited a review paper listed in reference [1] as the source to prove the scarcity of literature review on landscape assessment and perception. It could be interesting, if possible, to report quantitative information about the number of articles in this field.        

Author Response

See attached PDF

Reviewer 3 Report

Good paper.

Some explanations are needed especially in the first parts of the paper.

The reviewer understands that the author compares only 6 selected VIA methods.

Notes on VIA: 1. Approaches presented are predominantly subjective. 2. The results are related to existing KOPs and walking routes. 3. It would be appropriate to derive more pictorial documentation from the individual procedures assessed (I understand copyright issues).

4. More objective VIA methods are being developed and optimal KOPs are being sought for the relevant areas of the landscape later.

Many other VIA methods are available in Europe. See:

BELL, S. (1993): Elements of Visual Design in the Landscape. London, E & FN Spon, 212 p.

BRUNS, D., STEMMER B. (2018): Landscape Assessment in Germany. London, Routledge, 312 p.

DRAMSTAD, W. E., TVEIT, M. S., FJELLSTAD, W. J., FRY, G. L. A. (2006): Relationships between visual landscape preferences and map-based indicators of landscape structure. Landscape and Urban Planning, Vol. 78, pp. 465–474.

DUPONT, L., ANTROP, M., van EETVELDE, V. (2014): Eve-tracking Analysis in Landscape Perception Research: Influence of Photograph Properties and Landscape Characteristics. Landscape research, Vol. 39, pp. 417-432.

JANČURA, P. (2003): Charakteristický vzhľad krajiny – význam identifikácie vlastností krajinného obrazu a krajinného rázu v ochrane krajiny, plánovacích procesoch a hodnotení vizuálneho impaktu. Habilitačná práca. Banská Štiavnica, FEE TU, 88 p.

KUCHYŇKOVÁ, H., MIKITA, T. (2008): Visual Exposure Within the Dolni Morava Biosphere Reserve. Journal of Landscape Ecology, Vol. 1, pp. 67-79.

SALAŠOVÁ, A. (2020): Vizuálne hodnotenie krajiny v kontexte ochrany krajinného rázu. Životné prostredie, Vol. 54, pp. 131-141.

SEVENAT, M., ANTROP, M. (2010): The use of latent classes to identify individual differences in the importance of landscape dimensions for aesthetic preference. Land Use Policy, Vol. 27, pp. 827-842.

Author Response

See attached PDF
